| Open Peer Review | Biotechnology | Methods and Protocols

# Template plasmids optimized for deletion of multiple genes in yeast *Saccharomyces cerevisiae*

Yi-He Feng,[1] Jing-Zhen Song,[1] Jing Zhu,[1] Zhiping Xie[1]

**ABSTRACT** For *Saccharomyces cerevisiae*, gene knockout is routinely performed by transformation with a linear DNA cassette consisting of a selection marker gene flanked by upstream and downstream sequences homologous to a target gene. Over the years, several plasmid sets containing a variety of selection marker genes have been developed. Targeting fidelity under this strategy was high when performing the first gene knockout in a strain. However, we found that targeting fidelity decreased substantially when performing subsequent gene knockouts. The majority of the transformants were "incorrect," in which the new selection marker gene replaced a pre-existing selection marker gene instead of its intended target. This was caused by the presence of shared regions in the knockout DNA cassettes. To minimize shared regions among knockout cassettes, we developed a set of template plasmids, in which each selection marker open reading frame is flanked by a unique promoter/terminator combination. Our SJZ series templates cover eight selection markers, namely, *URA3* (*C. a.*), *TRP1* (*K.l.*), *his5* (*S.p.*), *LEU2* (*K.l.*), *nat*, *hph*, *kan*, and *amdS*. When using our templates, targeting fidelity in subsequent gene knockouts was restored to as high as that of the first knockout, with essentially all the transformants being correct. Our templates can therefore bring efficiency improvements in future research projects involving multi-gene knockouts.

**IMPORTANCE** When knocking out multiple genes in yeast, recombination among selection markers produces a large portion of false-positive transformants. We developed a new set of templates designed to minimize shared regions among selection markers. The use of this new template set resulted in essentially all transformants being correct knockouts.

**KEYWORDS** yeast, knockout, homologous recombination, selection marker, plasmid

Gene knockout is a routine procedure in many yeast research. Benefiting from high homologous recombination activity, homologous regions as short as 40 bp at each end of a linear DNA cassette are often sufficient for targeting a particular genomic locus. As a result, a typical knockout DNA cassette consisting of a selection marker gene flanked by upstream and downstream sequences homologous to a target gene can be constructed conveniently by the PCR amplification of a selection marker gene from a template plasmid using primer oligos containing short homologous sequences (Fig. 1A).

Since the 90s, several series of template plasmids have been developed, including the FA6a, UG, and AG series (Fig. 1A) (1–6). Their popularity in the yeast research field has continued until this day, as evidenced by the continued citation of the original publications (Fig. 1B). These plasmids cover a variety of selection markers. *URA3*, *TRP1*, *HIS3*, *LEU2*, and some of their fungal homologs can be used to complement corresponding auxotrophies (1–5, 7). As a result, their use requires the presence of auxotrophies in the parental strain. *kan*, *hph*, *nat*, and their derivatives enable antibiotic resistance to G418, hygromycin, and nourseothricin (1, 3, 4). *amdS* enables the use of acetamide as the sole

Address correspondence to Zhiping Xie, zxie@sjtu.edu.cn.

The authors declare no conflict of interest.

See the funding table on p. 8.

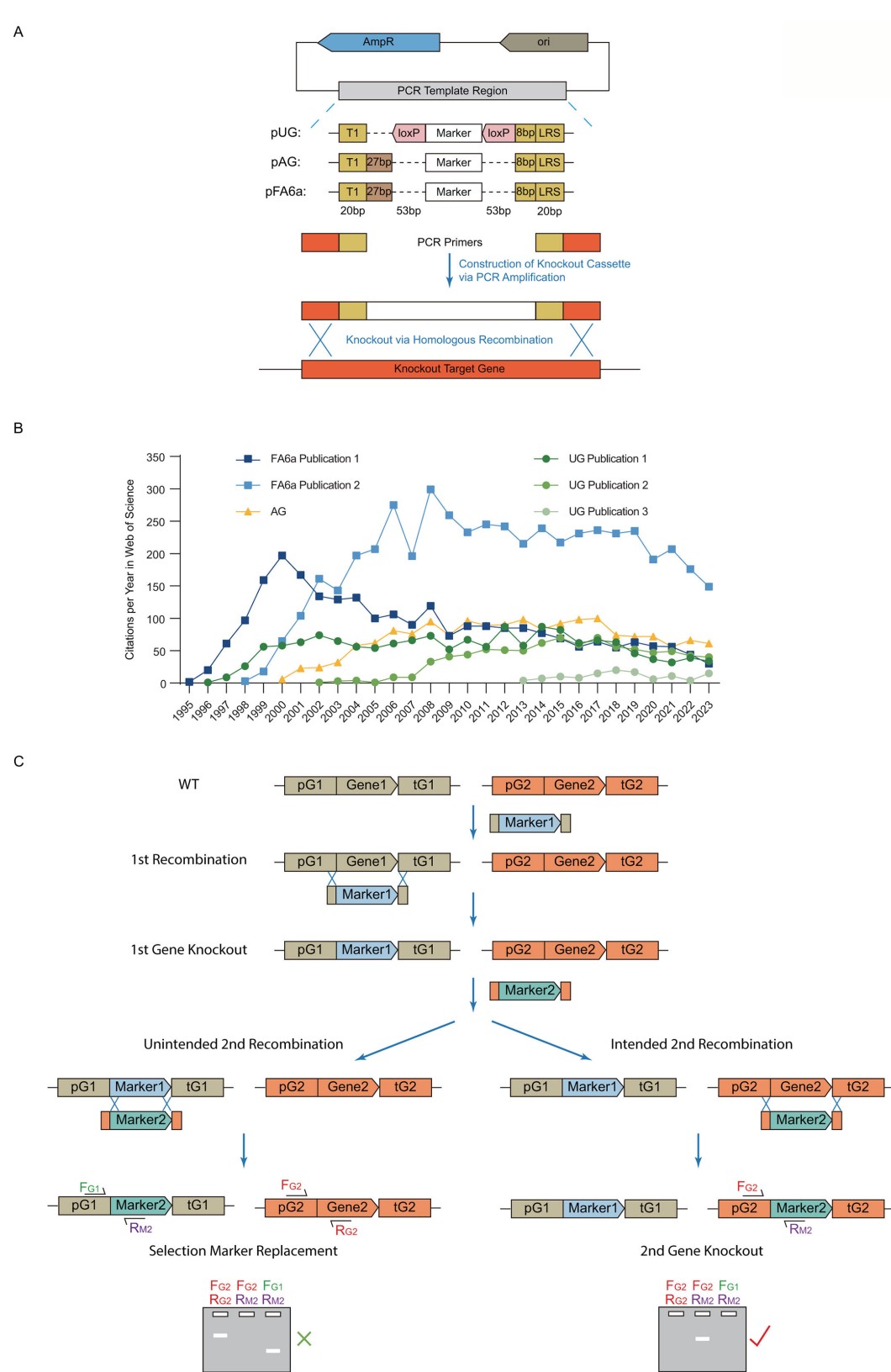

**FIG 1** General process of yeast gene knockout and issue of selection marker replacement in constructing multiple gene knockouts. (A) Construction of a DNA cassette for yeast gene knockout via recombination. A DNA knockout cassette containing a selection marker gene flanked by upstream and downstream (Continued on next page)

Fig 1 (Continued)

sequences targeting a particular gene can be constructed via PCR amplification of a template plasmid carrying the selection marker, with upstream and downstream targeting sequences introduced by the amplification primers. FA6a, AG, and UG series plasmids are commonly used templates. The three series share certain designs and allow the use of the same amplification primers. (B) FA6a, AG, and UG template series are actively used by the research community. Shown are the numbers of Web of Science citations per year for the original publications describing the FA6a (two paper), AG (one paper), and UG (three paper) series. (C) Unintended recombination of newly introduced cassette with a pre-existing selection marker instead of the intended knockout target. Hypothetical colony PCR results for incorrect and correct transformants are shown at the bottom.

nitrogen source (6). The latter four selection markers are particularly useful since their use requires no prior engineering of the parental strain.

The popularity of the FA6a, UG, and AG plasmids comes in part from the fact that, for any gene, a single pair of primer oligos can be used to construct a knockout cassette with any of the available selection markers (Fig. 1A). It is very convenient from a logistical perspective. For years, we have used a primer design strategy that allows the use of any of the FA6a, UG, and AG templates (Fig. 1A) and gradually came to the realization that the existing design has a limitation. The presence of shared regions among knockout cassettes constructed in this way substantially reduced targeting fidelity during the construction of multiple gene knockouts, even though the first knockout was highly accurate. The reduction was essentially unavoidable with the use of *his5(S.p.) kan*, *hph*, *nat*, and *amdS* since they all share a single promoter/terminator pair (Fig. 1A). The low targeting fidelity necessitates the testing of more transformants than what would be needed if fidelity was high. Methods are available to address this issue, but the methods all require more effort (see Discussion).

In the present work, we first use the UG series templates to demonstrate the phenomenon of reduced targeting fidelity during the knockout of multiple genes. We then describe the design of a new template set that aims to minimize the presence of shared regions among knockout cassettes and demonstrate the restoration of targeting fidelity with the use of our new templates.

## RESULTS

### "Incorrect" targeting of knockout cassettes to pre-existing selection marker genes

In FA6a, UG, and AG plasmid series, common stretches of sequences flank all the selection markers (Fig. 1A). When constructing a knockout cassette, we employed the established scheme of primer design with the first 40 nt of nucleotides matching upstream/downstream segments of a target gene, followed by 20 nt of nucleotide matching the common sequences on template plasmids. The two 20 bp segments we picked (T1 and LRS) avoided low-complexity regions. In this way, for any target gene, a knockout cassette containing any of the available selection markers can be amplified using a single pair of primers.

To demonstrate the phenomenon of "incorrect" targeting in the presence of pre-existing selection markers (Fig. 1C), we first constructed several knockout strains containing single- or double-gene knockouts. Note that even though we used the UG series as templates for this demonstration (Fig. 2A), the "incorrect" targeting phenomenon is not specific to the UG series. We then transformed these strains with knockout cassettes targeting additional genes (Fig. 2B). In the four cases (A1–A4), where newly introduced knockout cassettes carried marker genes (*hph*, *kan*, *nat*) sharing the *Ashbya gossypii TEF2* promoter/terminator with pre-existing selection markers [*nat*, *his5 (S.p.)*, *kan*], colony PCR verification revealed that success knockout rates ranged from 14 to 52% (Fig. 2B through F). In all the "incorrect" transformants, the newly introduced knockout cassettes replaced pre-existing selection markers instead of the intended targets (Fig. 1C and 2C through F). In cases (A2) and (A3), where the starting strains contained a mixture of selection markers with the *TEF2* promoter/terminator and those without, there was a preference toward replacing the pre-existing markers with the *TEF2* sequences (Fig. 2D

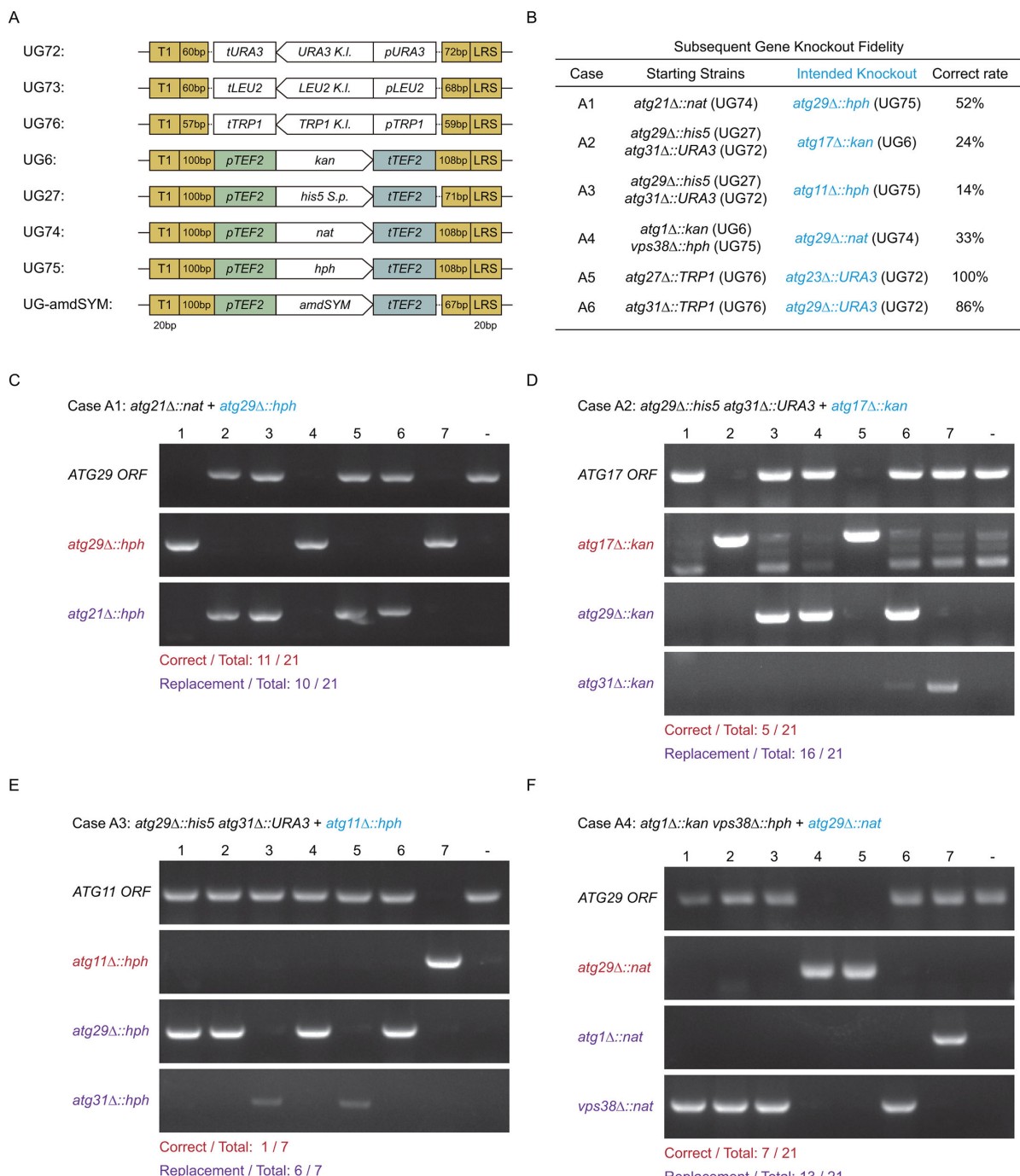

**FIG 2** Prevalence of selection marker gene replacement demonstrated with UG series templates. (A) Presence of shared regions in the amplified regions of UG series plasmids. T1 and LRS are regions where the amplification primers anneal to. *P* and t denote promoters and terminators, respectively. (B–F) Knockout fidelity was decreased in the presence of pre-existing knockout(s). Six cases were tested. In each case, the starting strain contained one or two genes knocked out using UG-based cassettes. A subsequent knockout of a different gene was performed using UG-based cassettes. Fidelity in this subsequent knockout operation was assessed by colony PCRs examining the status of intended knockout target and the status of pre-existing selection markers (see Fig. 1C for schemes of verification primer design). (B) Summary of knockout fidelity. (C–F) Agarose electrophoresis results of colony PCRs for cases A1–A4. 1–7, seven of the tested colonies. -, parental strain. Classification of all tested transformants (correct knockout vs replacement of a pre-existing marker) is listed below the gel images.

and E). Furthermore, in cases (A5) and (A6), where the newly introduced cassette (*URA3*) did not share promoter/terminator with the pre-existing selection marker (*TRP1*), rates of correct knockout were high (100 and 84%, respectively) (Fig. 2B). These data suggest that

the presence of shared regions among knockout cassettes is the cause of the "incorrect" targeting of newly introduced knockout cassettes.

## A new set of template plasmids minimizing shared regions among knockout cassettes

To improve targeting fidelity when constructing additional gene knockouts, we designed a new set of template plasmids (Fig. 3A). In this SJZ series of plasmids, the template regions to be amplified adhere to the following scheme: the first D segment (D1–D10, 51 bp, one in each template) and the second multiple-cloning-site (MCS, 35 bp) segment are both reserved for future development of this series into epitope-tagging templates. At present, either segment can be used as the template-annealing region in one of the PCR primers. The third segment contains the selection markers. The fourth and last segment is the R region (26 bp) containing the LRS region for annealing with the other PCR primer.

In our design, the open reading frame (ORF) of a selection marker is flanked by a promoter and a terminator from different genes, and the promoter/terminator combination is unique for each selection marker gene (Fig. 3A). This design has two primary purposes (see also Discussion). One is that, because the combination is unique to each selection marker, a newly introduced knockout cassette will not match a pre-existing one at both the promoter and the terminator, thus avoiding the aforementioned marker replacement issue. The second is that, with the promoter and the terminator in each pair derived from different genes, the recombination of knockout cassettes with the corresponding two genomic loci is unlikely to generate viable cells. We picked *Saccharomyces cerevisiae* (*S.c.*) promoters using the expression levels of the corresponding proteins as a guide (Fig. S1A) (8). The same strong *A.g. TEF2* promoter drives the expression of *kan*, *his5* (*S.p.*), *nat*, *hph*, and *amdS* in the UG series (Fig. 2A). Our use of high-level expression promoters from *FBA1*, *TEF1*, and *ENO2* to drive *nat*, *his5* (*S.p.*), and *amdS* worked well. However, for *hph* and *kan*, our initial use of high-level promoters from *TDH3* and *PGK1* resulted in the appearance of lawns on selection plates. Changing to promoters with about one-third of the strength, from *SSA1* and *GPM1*, allowed the appearance of individual colonies for these two dominant markers. For *URA3* (*K.l.*) and *TRP1* (*K.l.*), we picked promoters from *RPP1B* and *BBC1*, genes that are expressed several times higher than endogenous *URA3* and *TRP1*. We used the *LEU2* promoter to drive *LEU2* (*K.l.*). Other than avoiding matching and repetition, our choice of terminators did not involve special considerations. For ORFs, we choose *URA3* (*C.a.*) over *URA3* (*K.l.*) because, in our experience, the use of *URA3* (*K.l.*) generates colonies with drastically different sizes on selection plates.

## SJZ series templates afford high knockout fidelity for both the first gene knockout and subsequent gene knockouts

Using our SJZ series as the PCR template to generate knockout cassettes, we observed high targeting fidelity when knocking out the first gene (Fig. 3B). We tested seven transformants each in 13 knockout cases. Altogether, 89 out of 91 transformants were correct. We then tested the performance of our SJZ templates for gene knockouts in strains carrying pre-existing knockouts (Fig. 3C). Repeating the corresponding tests matching the A1–A4 cases (i.e., same starting strain, same target gene, and same selection marker ORF, except for using an SJZ plasmid as template), we obtained 26 correct ones out of 28 tested transformants (Fig. 3C through G). Notably, the two incorrect transformants in case B2 were not the result of selection marker replacement (Fig. 3F). The two might have experienced gene duplication. In test cases B5 and B6, matching cases A5 and A6, and in three additional test cases where the first gene was knocked out using our new SJZ templates, all tested transformants were correct. These data demonstrate a clear improvement of knockout fidelity by the use of SJZ series templates (Fig. 3D).

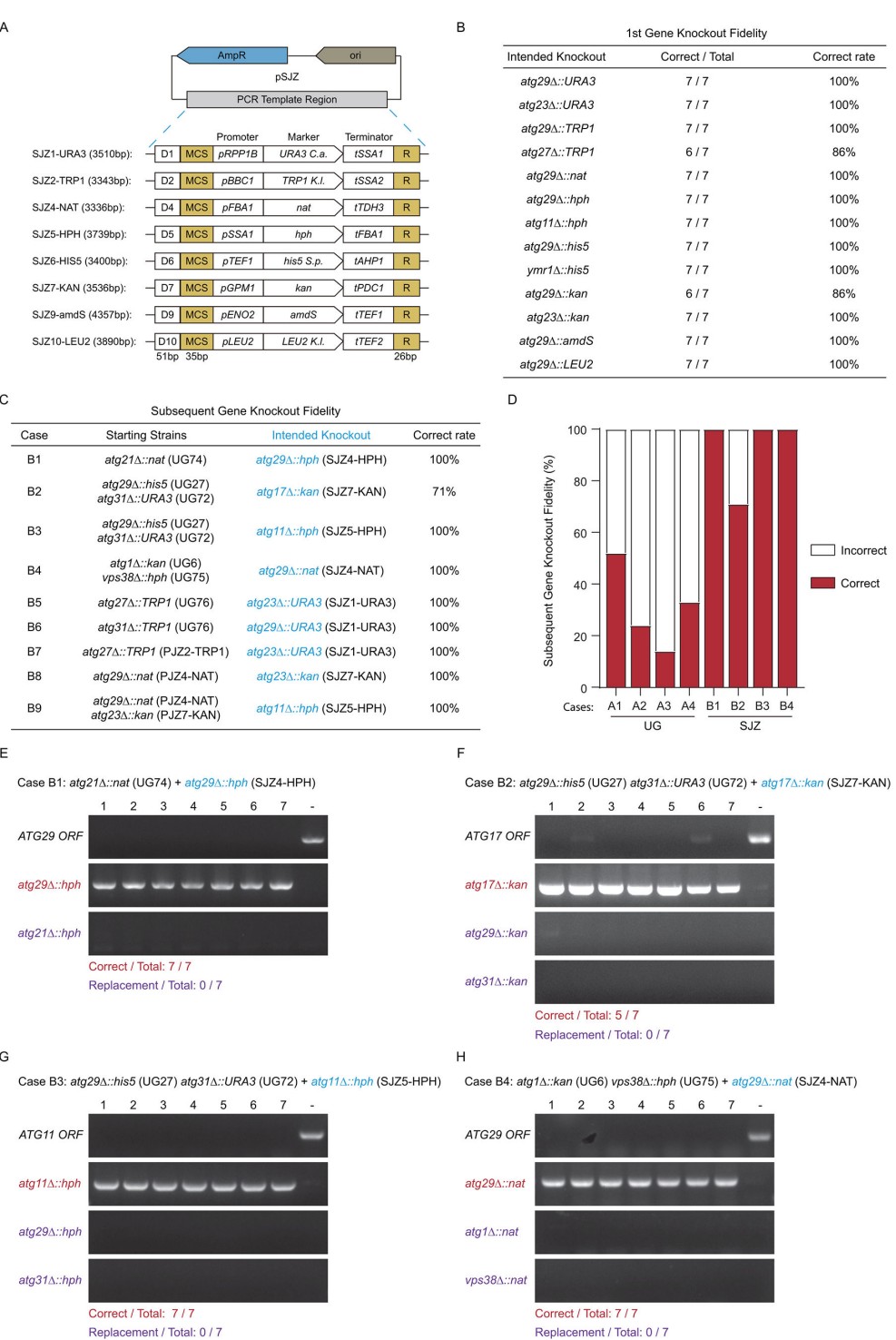

**FIG 3** Design of SJZ template series and restoration of knockout fidelity with its use. (A) Design of SJZ template series. A key feature is the use of unique promoter/terminator combinations for each selection marker open reading frame. Other non-essential shared regions are minimized. See main text for details. (B) Fidelity was high when knocking out the first gene using cassettes based on SJZ series. Thirteen knockout cases were tested. Results of colony PCR verification are summarized. (C–H) Fidelity was high when performing additional gene knockout using SJZ-based cassette in strains containing pre-existing knockouts using either UG- or SJZ-based cassettes. Nine cases were tested. The first six, B1–B6, matched cases A1–A6, except that the newly introduced knockout cassettes were based on SJZ templates. In the last three, B7–B9, the pre-existing knockouts were constructed using SJZ-based cassettes. (C) Summary of knockout fidelity. (D) Comparison of targeting fidelities between UG- and SJZ-based cassettes. (E–H) Agarose electrophoresis results of colony PCRs for cases B1–B4. Data presented as in Fig. 2C.

## SJZ series templates may also work for other yeast species

Many genetic elements developed for use in *S.c.* can also work in other yeast species. To see how far our SJZ series may work outside of *Saccharomyces*, we tested its use in *Schizosaccharomyces pombe* (*S.p.*). *S.c.* and *S.p.* belong to the same division, Ascomycota, but different classes (*Saccharomycetes* and *Schizosaccharomycetes*, respectively). For the four *S.c.* auxotropic markers, *URA3*, *TRP1*, *HIS3*, and *LEU2*, homologous genes are present in *S.p.* (Fig. S1B). However, *S.p.* Trp1 represents a merger of enzyme domains from *S.c.* Trp1 and Trp3 (9), precluding a direct use of our Trp1 marker in *S.p.* We, therefore, tested the use of four dominant markers and the remaining three auxotrophic markers for gene knockout in *S.p.* We constructed linear DNA flanked by approximately 300 bp of homologous targeting sequences (Fig. S1C) and found that those containing our *nat*, *hph*, *kan*, *LEU2*, and *his5* (*S.p.*) markers allowed the recovery of correct transformants (Fig. S1D through I). Knockout fidelities of *LEU2* and *his5* (*S.p.*) were not as high as what we observed in *S.c.* (Fig. S1D, compared with Fig. 3C). When using the *URA3* (*C.a.*) template, the growth of colonies was extremely slow on the initial selection plates. When using the *amdS* template, lawns were formed on the initial selection plates. The use of *URA3* (*C.a.*) and *amdS* markers may require additional optimizations and were not pursued further. These data suggest that five of our SJZ series templates can be used for gene knockout in *S.p.* without modification.

## DISCUSSION

The primary goal of developing the SJZ template series was to improve efficiency. Without the SJZ series, the issue of knockout cassette mistargeting can be solved by other methods. One is to use media recipe that selects for the presence of all the selection markers. In this approach, instead of preparing less than 10 types of selection media for single markers, a large variety of selection media is needed for each combination of selection markers, substantially increasing logistical complexity for a research lab. Furthermore, when using antibiotics, their concentrations when used in combination may need strain-dependent re-optimization. The other approach, which is the intended usage of the UG template series (4–6), is to use the Cre-loxP system to remove the pre-existing selection markers. The limitation of this approach is the investment of additional time, and the step needs to be performed for every parental strain. In contrast, with the high targeting fidelity from the SJZ templates, the extra effort of preparing multi-selection media varieties and the Cre-loxP operation are not needed. If one always tests four or more independent transformants during phenotype characterization, even the PCR verification of the transformants can be skipped, since one can assume that the majority of transformants will be correct.

Another usage of the SJZ series plasmids is the construction of experimental control strains. The parental strain without the new knockout cassette is not the ideal experimental control. In projects involving precise phenotype quantifications, the effects of introducing the selection markers need to be taken into consideration (10). By our design, each selection marker ORF is flanked by at least one segment of DNA (promoter or terminator) matching the corresponding genomic locus. Therefore, a selection marker can be introduced into the parental strain by transforming yeast with the template plasmid linearized at such promoter or terminator region, allowing the construction of a control strain carrying the selection marker gene with no additional gene knockout.

In summary, we demonstrated the presence of mistargeting issue during the construction of multiple gene knockouts, developed a new template set to address this issue, and demonstrated that our new tool served its purpose. The availability of the SJZ template set can bring cost and efficiency benefits to the yeast research community in future projects involving the construction of multiple gene knockouts.

## MATERIALS AND METHODS

### Construction of plasmids

The details of plasmid construction processes are listed in Table S1. Fragments were assembled via enzyme-mediated recombination. Reagents used included recombination enzymes (Vazyme C112, C113), DNA polymerase (Vazyme P501), and SacI restriction enzyme (New England Biolabs).

### Yeast selection media

YPD + NAT: YPD (1% yeast extract, 2% peptone, 2% glucose), 100 mg/L nourseothricin. YPD + KAN: YPD, 500 mg/L G418. YPD + HYG: YPD, 300 mg/L hygromycin B; SDN + amdS, 0.17% yeast nitrogen base (YNB) without amino acids and ammonium sulfate, 2% glucose, 0.6% acetamide; SMD dropouts: 0.67% YNB without amino acids, 2% glucose, amino acids and nucleotides (30 mg/L adenine, 20 mg/L histidine, 50 mg/L leucine, 50 mg/L tryptophan, 30 mg/L lysine, 20 mg/L uracil, 30 mg/L methionine, items dropped out as needed). In all cases, 2% agar was included for agar plates.

### Yeast transformation

Yeast cells were first cultured in a liquid YPD medium at 30°C. About 4 optical density (OD, 600 nm) log-phase cells were harvested and washed with LiTE (10 mM Tris pH 7.5, 1 mM EDTA, 100 mM lithium acetate). Cells were re-suspended in a composition containing 320 µL LiTE/PEG (10 mM Tris pH 7.5, 1 mM EDTA, 35% PEG4000, 100 mM Lithium acetate), 20–40 µL PCR-amplified knockout cassette, and 20 µL 5% ssDNA (Sigma D1626, dissolved in boiling water and frozen immediately) and undergone 45 min of heat shock at 42°C. Cells were washed with TE (10 mM Tris pH 7.5, 1 mM EDTA), and then cultured in a liquid YPD medium for 2 h. Finally, cells were washed with water and spread on an agar plate containing the appropriate selection medium.

Primers for the construction of knockout cassettes and colony verification are listed in Table S2. Strains used in this work are listed in Table S3.

## ACKNOWLEDGMENTS

The authors thank Dr. Li-Lin Du (National Institute of Biological Sciences, Beijing, China) for gifts of S.p. strains and members of Xie Lab for helpful discussions.

This work was supported by the National Key R&D Program of China (2020YFA0907700), the National Natural Science Foundation of China (32270796), and the Shanghai Municipal Science and Technology Commission (22ZR1433800).

Conceptualization, J.-Z.S.; Investigation, Y.-H.F. and J.-Z.S.; Writing—Original Draft, J.-Z.S. and Z.X.; Supervision, Z.X.; Project Administration, J.Z.; Funding Acquisition, Z.X.

## AUTHOR AFFILIATION

[1]State Key Laboratory of Microbial Metabolism and Joint International Research Laboratory of Metabolic & Developmental Sciences, School of Life Sciences and Biotechnology, Shanghai Jiao Tong University, Shanghai, China

## AUTHOR ORCIDs

Zhiping Xie http://orcid.org/0000-0001-5816-6159

## FUNDING

| Funder | Grant(s) | Author(s) |
| --- | --- | --- |
| MOST | National Key Research and Development Program of China (NKPs) | 2020YFA0907700 | Zhiping Xie |
| National Natural Science Foundation of China (NSFC) | 32270796 | Zhiping Xie |

| Funder | Grant(s) | Author(s) |
|---|---|---|
| Science and Technology Commission of Shanghai Municipality (STCSM) | 22ZR1433800 | Zhiping Xie |

## AUTHOR CONTRIBUTIONS

Yi-He Feng, Investigation | Jing-Zhen Song, Conceptualization, Investigation | Jing Zhu, Project administration | Zhiping Xie, Supervision, Writing – original draft

## ADDITIONAL FILES

The following material is available online.

### Supplemental Material

**Supplemental figure and tables (Spectrum01320-24-s0001.pdf).** Fig. S1; Tables S1 to S3.

### Open Peer Review

**PEER REVIEW HISTORY (review-history.pdf).** An accounting of the reviewer comments and feedback.

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
