## [Reviewer comments · Microbiology Spectrum]

Microbiology Spectrum

Template plasmids optimized for deletion of multiple genes in yeast *Saccharomyces cerevisiae*

Yi-He Feng, Jing-Zhen Song, Jing Zhu, and Zhiping Xie

Corresponding Author(s): Zhiping Xie, Shanghai Jiao Tong University

Review Timeline:

Submission Date:	May 30, 2024
Editorial Decision:	July 17, 2024
Revision Received:	September 15, 2024
Accepted:	September 20, 2024

Editor: Robert Arkowitz

Reviewer(s): The reviewers have opted to remain anonymous.

Transaction Report:

DOI: <https://doi.org/10.1128/spectrum.01320-24>

Re: Spectrum01320-24 (Template plasmids optimized for deletion of multiple genes in yeast *Saccharomyces cerevisiae*)

Dear Dr. Zhiping Xie:

Thank you for the privilege of reviewing your work. Below you will find my comments, instructions from the Spectrum editorial office, and the reviewer comments.

Revision Guidelines

Sincerely,

Robert Arkowitz
Editor
Microbiology Spectrum

Reviewer #1 (Comments for the Author):

To minimize shared regions among knockout cassettes, this study developed a set of template plasmids in which each selection marker ORF is flanked by a unique promoter/terminator combination. The experimental designs are sound and the manuscript is well structured. Please address the minor comments of reviewer #2.

Reviewer #2 (Comments for the Author):

The deletion cassettes designed by Xie et al improved the multiple gene deletion efficiency. The work is well done and useful for different yeast work labs. And it will be nice for sharing the cassettes in public database. Minor comments about the work are:

1. Line 134-139: For promoter and terminator selection in the vector construction, explain more why select these ones. Since normally strong constitutive promoters were used for the purpose of high expression of selection markers in yeast. Same as the selection of terminator, please explain more.
2. Whether the cassettes can be properly used in other yeast, like *S. pombe* or *C. glabrata*?

Responses:

We thank the reviewers for the careful evaluation of our manuscript and for the helpful suggestions. Following the suggestions, we revised our manuscript and included new experimental data. Please see details below.

Reviewer #1 (Comments for the Author):

To minimize shared regions among knockout cassettes, this study developed a set of template plasmids in which each selection marker ORF is flanked by a unique promoter/terminator combination. The experimental designs are sound and the manuscript is well structured. Please address the minor comments of reviewer #2.

Reviewer #2 (Comments for the Author):

The deletion cassettes designed by Xie et al improved the multiple gene deletion efficiency. The work is well done and useful for different yeast work labs. And it will be nice for sharing the cassettes in public database. Minor comments about the work are:

1. Line 134-139: For promoter and terminator selection in the vector construction, explain more why select these ones. Since normally strong constitutive promoters were used for the purpose of high expression of selection markers in yeast. Same as the selection of terminator, please explain more.

Responses:

We initially followed the same rationale as stated by this reviewer in using stronger promoters. It worked well for nat, his5, and amdS. However, lawns were formed when testing hph and kan. We switched to lower strength promoters for hph and kan, and got individual colonies. For URA3, TRP1, and LEU2, we used promoters either several folds higher than endogenous or the endogenous one. No special consideration was taken for terminator choice other than the originally stated principle of avoiding mistargeting.

This information is now included in the main text in Page 5 Line 119 to Page 6 Line130.

2. Whether the cassettes can be properly used in other yeast, like *S. pombe* or *C. glabrata*?

Responses:

This is a very important question. Since *C.g.* is more closely related to *S.c.* than *S.p.*, we decided to test the use of our SJZ templates in *S.p.* to see how far we can possibly go. Among the eight markers, *S.c.* TRP1 cannot substitute *S.p.* trp1 for theoretical reasons. We tested the remaining seven and found that five worked in *S.p.* (nat, hph, kan, LEU2, and his5). It is possible that the other two (URA3 and amdS) need *S.p.* specific optimizations. The new *S.p.* data are presented in

Fig. S1, and described in the main text in Page 6 Line 149 to Page 7 Line 166.

Re: Spectrum01320-24R1 (Template plasmids optimized for deletion of multiple genes in yeast *Saccharomyces cerevisiae*)

Dear Dr. Zhiping Xie:

Thank you for addressing the reviewers comments.

Your manuscript has been accepted, and I am forwarding it to the ASM production staff for publication. Your paper will first be checked to make sure all elements meet the technical requirements. ASM staff will contact you if anything needs to be revised before copyediting and production can begin. Otherwise, you will be notified when your proofs are ready to be viewed.

Sincerely,
Robert Arkowitz
Editor
Microbiology Spectrum